# Socio-Demographic Factors and Public Knowledge of Antibiotic Resistance

**DOI:** 10.3390/healthcare11162284

**Published:** 2023-08-14

**Authors:** Vedika Bhatt, Sasheela Sri La Sri Ponnampalavanar, Chun Wie Chong, Li Yoong Tang, Karthikayini Krishnasamy, Sheron Sir Loon Goh, Cindy Shuan Ju Teh

**Affiliations:** 1Department of Medical Microbiology, Faculty of Medicine, Universiti Malaya, Kuala Lumpur 50603, Malaysia; vedikabhatt@um.edu.my; 2Department of Medicine, Faculty of Medicine, Universiti Malaya, Kuala Lumpur 50603, Malaysia; 3School of Pharmacy, Monash University Malaysia, Bandar Sunway 47500, Selangor, Malaysia; chong.chunwie@monash.edu; 4Department of Nursing Science, Faculty of Medicine, Universiti Malaya, Kuala Lumpur 50603, Malaysia; lytang@ummc.edu.my; 5Department of Nursing, Universiti Malaya Medical Center, Kuala Lumpur 50603, Malaysia; karthikayini@ummc.edu.my; 6Department of Primary Care Medicine, Faculty of Medicine, Universiti Malaya, Kuala Lumpur 50603, Malaysia; sherongoh@yahoo.com

**Keywords:** antibiotic, antibiotic-resistance, knowledge, awareness, Malaysia

## Abstract

(1) Background: Antibiotic resistance is a serious health issue, and raising public awareness of it is crucial to combating it. This study aimed to assess the socio-demographic factors associated with knowledge of antibiotics and antibiotic resistance in Malaysia. (2) Methods: A cross-sectional study was carried out between April 2022 and March 2023. Malaysian adults aged ≥18 years old and able to understand English or Malay were recruited. During data collection, the WHO questionnaire “Antibiotic Resistance, Multi-Country Public Awareness Survey” was used. Data were collected across 14 states in Malaysia. (3) Results: A total of 517 participants completed the questionnaire. Most participants were females (67.9%), aged 30–49 (46%), and from central Malaysia (69.8%). Most participants (98.5%) reported taking antibiotics. A misconception presented was that sore throats, fevers, colds, and flu can be treated with antibiotics. A total of 58.8% of participants had high knowledge of antibiotic usage (scores 12–15), while 64% had high knowledge of antibiotic resistance (scores 9–14). Findings indicate that increasing age, income, and education were associated with higher knowledge. (4) Conclusions: This study highlights the knowledge deficiency of antibiotic resistance among Malaysians. Educational programs should engage a younger and lower socio-economic population to increase awareness.

## 1. Introduction

Antibiotics typically function by binding to specific target sites of bacteria to disrupt their activity. However, throughout evolution, bacteria have developed resistance to antibiotics, causing the medication to be ineffective in fighting infections [1]. Increased antibiotic resistance is the cause of severe infections, complications, longer hospital stays, and increased mortality. It is also a financial burden as it leads to high healthcare expenses, especially due to the additional expenses of hospital resources and medications [2]. As a matter of fact, it was estimated that additional costs due to antibiotic resistance could range from USD 300 billion to more than USD 1 trillion a year by 2050 [3]. Antibiotic resistance rises and falls in a cyclic manner, increasing with the misuse and overuse of antibiotics and decreasing with their disuse [4]. Overprescribing antibiotics is a particular issue in primary care, where viruses cause most infections. About 90% of all antibiotic prescriptions issued by general practitioners for respiratory tract infections were unnecessary and became the leading cause of overprescribing [5].

The threat of antibiotic resistance is worldwide. The estimated global mortality rate from drug resistance in 2019 was 4.95 million, with Southeast Asia accounting for over a thousand deaths [6]. The Ministry of Health in Malaysia has been actively carrying out surveillance of government hospitals. According to their 2021 report, the rates of ESBL-producing *Klebsiella pneumoniae*, *Acinetobacter baumannii*, Carbapenem-Resistant Enterobacteriaceae (CRE), and Vancomycin-Resistant Enterococci (VRE) have been increasing, while the rates of Methicillin-resistant Staphylococcus aureus show no change from previous years. A total of 23% of the clinical isolates reported came from the community, while 77% came from healthcare-associated infections [7]. In terms of antibiotic usage, an increase in the use of carbapenems, piperacillin/tazobactam, and total polypeptides was reported in Malaysian hospitals [8]. There is not sufficient data on the actual antibiotic consumption in private hospitals and clinics. Nonetheless, certain write-ups, such as the National Medical Care Survey (NMCS), indicate private clinics were the major contributors to antibiotic prescriptions among all primary care antibiotic prescriptions [9].

As antibiotic misuse is the cause of antibiotic resistance, there is an urgent need to raise public awareness regarding antibiotic misuse. A previous systematic review found that the public has an incomplete understanding of antibiotic resistance and its causes. Even though 70% of participants were familiar with the term “antibiotic resistance,” the majority (76%) thought it meant changes in the human body [10]. Previous studies across Asia found that 30–70% believed viral infections respond to antibiotics and 60–83% lacked understanding of how antibiotic resistance develops [11,12,13,14]. Similar studies have revealed that Malaysians have less to moderate knowledge about using antibiotics and antibiotic resistance, and almost half of the participants had not finished their course of antibiotics [15,16,17]. To combat the emergence of antibiotic resistance, the WHO developed a Global Action Plan (GAP) to increase understanding of antibiotic resistance [18]. In Malaysia, multiple initiatives like the Malaysian Action Plan on Antimicrobial Resistance (MyAP-AMR) 2017–2021 and the World Antibiotic Awareness Week have been developed to slow down and prevent the emergence of antibiotic resistance [19]. These initiatives could increase public awareness about antibiotic usage and antibiotic resistance [20]. However, to date, a recent national-scale investigation on the usage and knowledge of antibiotics has not been carried out. Therefore, this study aimed to identify the association between different socio-demographic characteristics and the level of knowledge of antibiotics and antibiotic resistance.

## 2. Materials and Methods

### 2.1. Study Setting and Population

This study was a cross-sectional study carried out to assess the public’s knowledge of antibiotics and antibiotic resistance in Malaysia for a span of 11 months, from April 2022 to March 2023. The WHO questionnaire, “Antibiotic Resistance, Multi-Country Public Awareness Survey,” was used for this study [10]. The questionnaire was available for Malaysian participants in English and Bahasa Melayu, which is the national language of Malaysia. The Bahasa Melayu language questionnaire was translated by a group of experts to ensure the quality of the translation. Google Forms was used as the platform for participants to answer the survey. Participants included were Malaysian adults aged 18 years old and above. Participants who did not understand English or Bahasa Melayu were excluded. The questionnaire was disseminated nationwide through email, social media, and phone. To improve inclusivity, surveys were also interviewer–administered to the public, including older people and people who were not well-versed in technology. All responses to the questionnaire were completely voluntary and anonymous.

The sample size was obtained using the Raosoft sample size calculator. The margin of error used was 5%, with a confidence level of 95% for a Malaysian population of 32.7 million in 2021 [21,22]. Hence, the minimum participant sample size for this study was set at 385 participants. The questionnaire was sent through email, social media, and word-of-mouth. This study was approved by the Universiti Malaya Medical Research Ethics Committee (UM MREC). The MREC ID is as follows: 2021910-10578.

### 2.2. Survey Questionnaire

The sections covered in the questionnaire consisted of section A, participant demographics (Q2–Q9); section B, use of antibiotics (Q10–Q13); section C, knowledge about antibiotics (Q14–Q17); section D, knowledge about antibiotic resistance (Q18–Q25); section E, participant opinions on antibiotic resistance (Q26–Q27); and section F, use of antibiotics in agriculture (Q28). All the questions used a closed-question methodology to maintain consistency in the answers provided. The demographics that were asked consisted of gender, age, residence, education, household income, ethnicity, and household composition. The survey consisted of multiple-choice questions with single-select, multiple-select answer options, and 5-point Likert scale questions. Participants’ names and personal identifiers were not requested to maintain anonymity. Before answering the survey, a summary of this study was presented to the participants. Consent was electronically obtained when participants clicked on “I consent” before starting the survey.

### 2.3. Data Analysis

Data were analyzed using the Statistical Package for Social Sciences (SPSS) version 29 (SPSS; Chicago, IL, USA). The data obtained were analyzed using descriptive analysis and Pearson’s chi-square or Fisher’s exact tests. A *p*-value of less than 0.1 was considered statistically significant. For sections C and D, scores were calculated to indicate the knowledge of antibiotics and antibiotic resistance among participants. A score of “1” was given for each correct answer and “0” for an incorrect answer. To categorize the scores of the participants, the median scores for the knowledge of antibiotics and the knowledge of antibiotic resistance were calculated, respectively. Participants that had a lower score than the median were grouped as having “low knowledge,” while those that had scores higher than the median were grouped as having “high knowledge.” To understand the association between demographic factors and the knowledge of participants, regression analysis was carried out. The degree of correlation between knowledge of antibiotics and knowledge of antibiotic resistance was analyzed using Spearman’s rank-order correlation coefficient.

## 3. Results

### 3.1. Participant Characteristics

During data collection, a total of 517 survey responses were obtained from participants around Malaysia. The socio-demographic data are shown in Table 1. There was a higher representation of females (n = 351, 67.9%), ages 30–49 (n = 238, 46.0), and those from urban settlements (n = 368, 71.2%). The majority of the participants also come from the central region of Malaysia, particularly from Selangor (n = 210, 40.6%) and Kuala Lumpur (n = 139, 26.9%).

In terms of education, most participants have pursued higher education, which includes having a Bachelor’s degree, Master’s degree, or Doctorate (n = 365, 70.6%). As for the monthly household income, responses were divided into three primary groups based on the Malaysian income classification, which are the below 40 group (poor income), the middle 40 group (average income), and the top 20 group (rich income) [23].

### 3.2. Use of Antibiotics

A large proportion (n = 509, 98.5%) recall taking antibiotics at some point in their life, while only 8 (1.5%) reported never having antibiotics. Out of those who have taken antibiotics before, 465 participants (91.4%) obtained a prescription and were also given directions on how the medication is to be taken. Most of these participants (95.1%) obtained their medication at a medical store or pharmacy. The data are shown in Table 2.

### 3.3. Knowledge of Antibiotics and Antibiotic Resistance

#### 3.3.1. Knowledge of Antibiotics

The scores in this section were categorized into the “0–11” group or the “12–15” group based on the median split, which was calculated to be 12.0. Most participants received a “high knowledge” score (n = 304, 58.8%), while 213 (41.2%) received a “low knowledge” score. Moreover, a better antibiotic knowledge level was noted in participants aged 30–49 (X^2^ = 33.464, *p* = 0.000), with rich household income (X^2^ = 9.700, *p* = 0.008), and those with higher education (X^2^ = 19.450, *p* = 0.000). Evaluation of the scores also revealed that only 42 participants (8.1%) obtained a complete score of 15. The association between the socio-demographics of participants and their level of knowledge of antibiotics is shown in Table 3.

A total of 38 participants (7.4%) wrongly think antibiotics should be stopped once they feel better, while 4 participants (0.8%) do not know. When asked if it is acceptable to take antibiotics given by friends or family, 22 participants agreed (4.3%), while 22 (4.3%) did not know. The misconception of buying or requesting an antibiotic when having similar symptoms as previous illnesses was noted, as 72 participants (13.9%) agree that this is acceptable, while 47 (9.1%) do not know. Participants were also asked about health conditions that can be treated with antibiotics, as shown in Figure 1. The most common illnesses selected were bladder or urinary tract infections (UTI) (n = 397, 76.8%) and skin or wound infections (n = 373, 72.1%). Participants wrongly identified sore throat (n = 231, 44.7%), fever (n = 191, 36.9%), cold and flu (n = 162, 31.3%), diarrhea (n = 123, 23.8%), and other illnesses (n = 173, 33.5%) as conditions treatable with antibiotics. In comparison, only 192 (37.1%) selected gonorrhea correctly.

#### 3.3.2. Knowledge of Antibiotic Resistance

Based on the median score of 9.0 in this section, a median split divided scores of “0–8” and “9–14” into the “low knowledge” and “high knowledge” groups, respectively. Most participants obtained a high score (n = 331, 64.0%), while 186 (36.0%) obtained a low score. Interestingly, there were no participants that obtained a full score of 14, and only one (0.2%) participant obtained the next highest score of 13. Participants of age 30–49 (X^2^ = 5.457, *p* = 0.065), male (X^2^ = 2.930, *p* = 0.087), rich income (X^2^ = 24.129, *p* = 0.000), and those living in the Southern region (X^2^ = 12.114, *p* = 0.015) received a high knowledge score. The association between the socio-demographics of participants and their level of knowledge of antibiotic resistance is shown in Table 3.

Six phrases that are frequently used when referring to the problem of antibiotic resistance were listed, and participants were asked if they had heard of them. Out of these phrases, “Drug Resistance” (19.6%) and Antibiotic Resistance” (19.6%) were most heard, with only a negligible difference in their values. This is followed by the phrases “Antibiotic-resistant Bacteria” (18.5%), “Superbugs” (15.6%), “Antimicrobial Resistance” (15.2%), and the abbreviation “AMR” (11.4%). A total of 63 participants (12.2%) have not heard any of these terms. Figure 2 shows the participants’ sources of information. Most individuals heard it from physicians or nurses (38.7%).

To investigate the knowledge level of participants about antibiotic resistance, participants were also provided with a set of statements and asked whether they were true or false. Although most of these statements were answered correctly, a common misunderstanding of how antibiotic resistance works was observed, as 451 participants (87.2%) wrongly agreed that the body becomes resistant to antibiotics, leading to antibiotic resistance.

#### 3.3.3. Factors Associated with Knowledge of Antibiotics and Knowledge of Antibiotic Resistance

A multivariate logistic regression analysis was performed to ascertain the effects of significant socio-demographic characteristics such as gender, age, race, household income, education level, and region on the likelihood of having a high knowledge of antibiotics and antibiotic resistance (Table 4). The unstandardized coefficient (B) from the analysis indicates that when age, household income, and education level increase, a higher tendency toward knowledge is present.

For “Knowledge of Antibiotics,” age and education level have a significant impact on the higher possibility of achieving a high knowledge level, as indicated by the adjusted odds ratio and confidence interval. As for “Knowledge of Antibiotic Resistance,” the socio-demographic characteristics of household income, education level, and geographical region indicate a higher tendency to have an outcome of high knowledge.

#### 3.3.4. Relationship between the Knowledge of Antibiotics and the Knowledge of Antibiotic Resistance

Spearman’s rank-order correlation indicated a weak positive monotonic correlation between knowledge of antibiotics and knowledge of antibiotic resistance (ρ=0.306). Nevertheless, the correlation is significant, as shown in Table 5.

### 3.4. Public Opinion on Antibiotic Resistance

In addition to obtaining an insight into the level of knowledge, a Likert scale was used to understand the different opinions the public has on actions that could potentially combat antibiotic resistance (Figure 3). Most responses indicated that the public knew the correct courses of action for addressing the issue of antibiotic resistance. A total of 508 participants (98.3%) agreed that parents should be responsible for making sure their children receive their vaccinations. The importance of hand hygiene was also highlighted, as 507 participants (98.1%) agreed that hands should be washed regularly. As for actions taken by non-healthcare practitioners such as pharmaceutical companies, governments, and farmers, a hint of uncertainty was displayed towards their roles in combating resistance as more participants were neutral or in disagreement.

Another Likert scale asked participants about the seriousness of antibiotic resistance and their role in the problem. A more diverse set of responses that indicate unfamiliarity with antibiotic resistance could be seen (Figure 4). More than half (54.7%) of the participants believed that they were not at risk of antibiotic-resistant infection if they took the medication as prescribed. Only 174 (33.7%) rightfully believed that they too could help stop antibiotic resistance. A total of 132 individuals (25.5%) did not fully agree that antibiotic resistance is one of the biggest problems facing the world.

### 3.5. Knowledge about Antibiotic Use in Agriculture

The final section of the survey asks participants if they think antibiotics are widely used in agriculture in Malaysia. A total of 324 (62.7%) participants agreed, 20 (3.9%) disagreed, and 173 (33.5%) were unsure about this statement.

## 4. Discussion

This cross-sectional study was able to recruit 517 participants from different parts of Malaysia. This overall response rate exceeded our expectations in providing important data on knowledge of antibiotic usage and resistance in Malaysia, as well as possibilities to decrease the misuse of antibiotics. Antibiotic resistance is a prevalent issue in Malaysia, as rates of antibiotic-resistant bacteria have shown a rise in resistance patterns [24]. A comparison that can be drawn is the difference in the usage of antibiotics globally. According to the Eurobarometer, Malta had the highest consumption of antibiotics over the past 12 months (42%), while Sweden had one of the lowest (15%) in 2020. Additionally, Sweden presented a higher knowledge level regarding knowledge of antibiotics compared to Malta [25]. On the contrary, our findings in Malaysia showed that 53% of people had taken antibiotics in the past 12 months. A global study on antibiotic usage also reported Malaysia to have one of the highest antibiotic consumptions, in contrast to other Southeast Asian countries with notably low usage of antibiotics, like Indonesia and the Philippines [6]. Nevertheless, our responses indicate that people still obtain antibiotics without a proper prescription, even though it is an illegal practice (4.7%). Although our results were better than those of other Asian countries such as Myanmar (58.6%), Indonesia (40%), and Thailand (29.7%), stricter inspections should be carried out on antibiotic dispensing [14,26,27].

Based on the results of this study, 41.2% have a low knowledge score on antibiotic usage. Most of the participants correctly answered that all antibiotics are to be taken as directed, even if symptoms subside (91.9%). This overall result is better compared to the previous studies carried out in Malaysian cities such as Pahang, Putrajaya, and Shah Alam [28,29]. As for antibiotic-treatable health conditions, most participants correctly identified bladder or urinary tract infections as well as skin or wound infections. However, it is important to highlight that participants also incorrectly selected sore throat, fever, cold and flu, and diarrhea in accordance with the responses of the multi-country report published by WHO [10]. Similar findings were also reported in the literature in Nigeria and European countries [25,30]. A possible reason for this could be because of the physician’s attitude. Despite being aware that antibiotics have limited use in certain medical conditions, there are physicians who continue to prescribe them. Patients’ expectations, the severity and length of illnesses, the uncertainty around the diagnosis, and the implemented policies are some of the factors that influence a physician’s prescribing [31].

Responses on knowledge of antibiotic resistance indicate that 36% have low knowledge. Consistent with results from other studies, “Drug Resistance” and “Antibiotic Resistance” are terms that people are most familiar with. A reason for this could be because these terms are more self-explanatory compared to others like “Superbugs” and “AMR.” Our data demonstrate that healthcare professionals have played a key role in informing patients about antibiotic resistance. This shows that the stringent efforts of the Malaysian government to implement policies and guidelines for infection control and prevention in healthcare settings have been successful [32]. Our findings also report that participants fail to acknowledge antibiotic resistance as a characteristic of the bacteria. Instead, they believe that developing resistance means that the body has grown accustomed to taking antibiotics and that they are no longer effective. This unchanging misapprehension has been stated in literature over the years [16,33]. Participants also incorrectly think that AMR is only a problem for those who frequently take antibiotics. This misunderstanding coincides with the responses that agree that bacteria that are resistant do not spread from person to person. The rapid spread of resistant organisms has been made simpler by modern transportation, as individuals can carry bacteria across and between continents [34].

Upon post-hoc analysis, a significant association was highlighted between socio-demographic characteristics and high knowledge levels. Education level was the only common factor associated with high knowledge of antibiotics and antibiotic resistance. This pattern is replicated in other studies conducted in Bhutan, Hong Kong, and Cyprus [13,34,35]. Tertiary education could be associated with higher knowledge as students have access to more resources. Another factor that was associated with high “Knowledge of Antibiotics” was the increase in the age of participants. This could stem from their increasing exposure to healthcare interactions and experiences compared to lower age groups. As for the level of “Knowledge of Antibiotic Resistance,” household income and geographical location also contribute to having high knowledge. According to Zailani et al., individuals with higher income levels would have access to better educational opportunities and healthcare services, causing them to have higher knowledge [36]. The positive correlation between “Knowledge of Antibiotics” and “Knowledge of Antibiotic Resistance” implies that people who are familiar with antibiotics are inclined to comprehend the associated subject of antibiotic resistance in greater depth.

Furthermore, there was a lack of understanding of the seriousness of antibiotic resistance and individual roles in combating it. Less than half (40.0%) of the participants agreed that there is not much that they can do to stop antibiotic resistance. The WHO survey to assess the knowledge and awareness of antibiotic resistance across 12 countries in 2015 also had similar findings, with 57% saying they do not have a role [10]. Preventing antibiotic resistance is not just the role of healthcare experts; it is a multidisciplinary effort. The public can especially prevent antibiotic resistance by educating themselves on the appropriate use of antibiotics, getting their vaccinations, advocating for policies surrounding the issue, and supporting sustainable agricultural practices. Some of the positive findings of the study indicate that Malaysians strongly understand the importance of washing hands and timely vaccinations. These results are comparable to those of a survey carried out in Singapore that indicated lower importance placed on hand hygiene and vaccinations [37]. Participants in this study were also unsure if new antibiotics should be developed. According to Lee et al., antibiotic development is crucial to ensuring we have solutions to fight against bacteria that are resistant to antibiotics [38]. Uncertainty is observed in responses to the question of whether farmers should give fewer antibiotics to food-producing animals. Research shows that decreasing on-farm antibiotic use appears to be the most effective strategy for addressing the issue of antibiotic resistance coming from animals in Malaysia, where antibiotic resistance is a severe health problem [15]. It is important for the public to know this, as they are consumers of food-producing animals.

Educational initiatives should investigate these knowledge gaps to develop a more apprehensive way to address this issue. There is evidence that targeted educational intervention raises awareness of antibiotic resistance, as presented by a local study in the state of Perak [39]. According to the study, knowledge of antibiotic use and antibiotic resistance has increased post-intervention. Another Malaysian study has shown that the Malaysian public has good knowledge of COVID-19 but low knowledge of antibiotic use and antibiotic resistance [40]. This could be due to the mass dissemination of information on COVID-19 during the pandemic. Similarly, with more attention to antibiotic resistance, public knowledge could be improved. As such, we plan to carry out future studies targeting the younger age group by carrying out qualitative studies on their understanding of antibiotic resistance as well as the development of training modules.

There are several limitations to this study. First, most participants were from the Malaysian metropolitan cities of Selangor and Kuala Lumpur. This may cause an underrepresentation of the whole Malaysian population. Second, there is a possibility of recall bias in answers such as household income and participants’ prior use of antibiotics. Third, it is difficult to monitor the understanding of participants as this study has been conducted using an online questionnaire due to the possible cost imposed by the geographical factors of an in-person survey. Nonetheless, an online questionnaire allows us to reach a larger group of respondents, as most Malaysians use smartphones. The anonymous nature of the questionnaire could encourage honesty because participants would not feel judged or self-conscious during the public awareness survey.

## 5. Conclusions

To our knowledge, this study is one of the first assessments of knowledge of antimicrobial resistance conducted on a national scale after COVID-19. Despite the high usage of antibiotics in the country, our results indicate that the knowledge of antibiotic use and antibiotic resistance among the Malaysian public is still poor. Younger individuals, as well as those with a low education level and low household income in particular, present lower knowledge. The gaps in knowledge highlighted in this study could be used to plan educational interventions to raise awareness about antibiotic resistance across the country. When the public understands their roles in combating antibiotic resistance and the seriousness of the issue, the spread of antibiotic-resistant infections could slow down. This could lessen the strain on the healthcare system and maintain the efficacy of already available antibiotics. To identify the underlying causes of human attitudes toward and perceptions of antibiotic resistance, a qualitative analysis is necessary.

## Figures and Tables

**Figure 1 healthcare-11-02284-f001:**
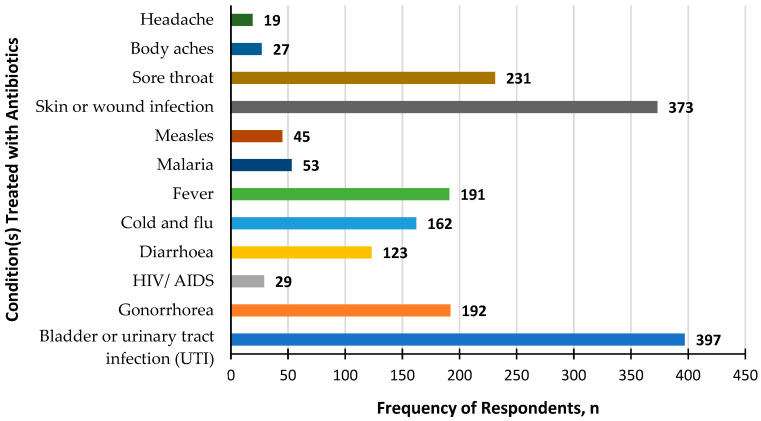
Health conditions can be treated with antibiotics, according to participants.

**Figure 2 healthcare-11-02284-f002:**
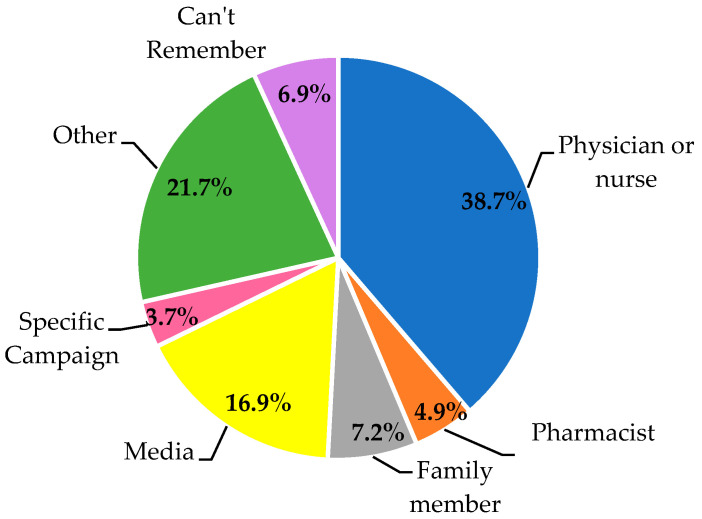
Sources of information about antibiotic resistance.

**Figure 3 healthcare-11-02284-f003:**
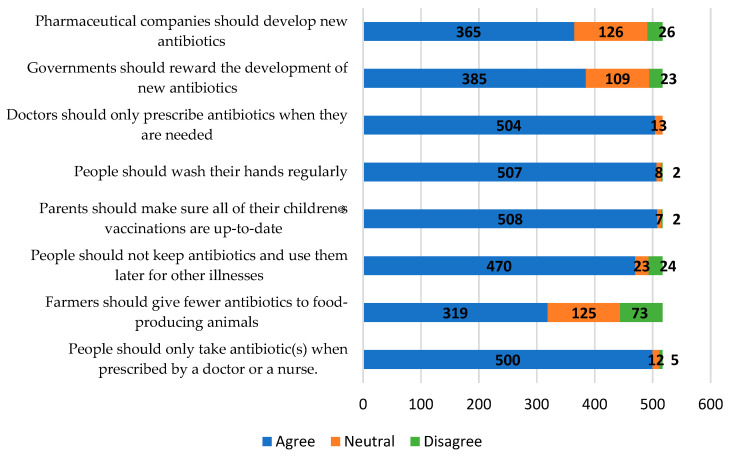
Participants’ opinions on ways of addressing the problem of antibiotic resistance.

**Figure 4 healthcare-11-02284-f004:**
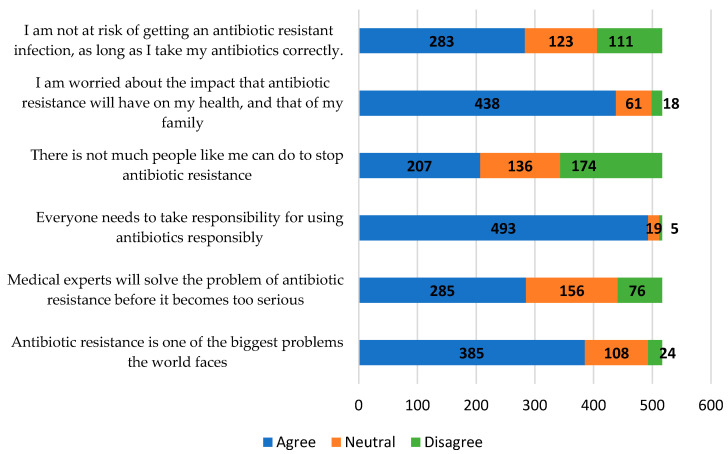
Participants’ opinions on the seriousness of antibiotic resistance.

**Table 1 healthcare-11-02284-t001:** Socio-demographic characteristics of the participants (n = 517).

Characteristics of Participants	Frequency, n = 517	Percent (%)
Gender		
Male	166	32.1
Female	351	67.9
Age (Years)		
18–29	148	28.6
30–49	238	46.0
50–60+	131	25.3
Region(s) in Malaysia		
Northern Region ^1^	71	13.7
Southern Region ^2^	37	7.2
Central Region ^3^	361	69.8
East Coast ^4^	7	1.4
Borneo Region ^5^	41	7.9
Settlement Category		
Urban	368	71.2
Suburban	133	25.7
Rural	16	3.1
Ethnicity		
Malay	153	29.6
Chinese	153	29.6
Indian	171	33.1
Other	40	7.7
Education Level		
High school and less	34	6.6
Post-secondary	118	22.8
Higher Education	365	70.6
Household Income		
Low Income	189	36.6
Average Income	186	36.0
High Income	142	27.5
Household Composition		
Adults	310	60.0
Adult and Children	207	40.0

^1^ Northern region: Perlis, Kedah, Penang, and Perak. ^2^ Southern region: Johor and Malacca. ^3^ Central region: Selangor, Kuala Lumpur, Putrajaya, and Negeri Sembilan. ^4^ East Coast: Terengganu, Kelantan, and Pahang. ^5.^ Borneo region: Sabah, Sarawak, and Labuan.

**Table 2 healthcare-11-02284-t002:** Usage of Antibiotics.

Use of Antibiotics	Frequency, n	Percent (%)
Last use of antibiotics (n = 517)		
In the last month	74	14.3
In the last 6 months	134	25.9
In the last year	66	12.8
More than a year ago	162	31.3
Cannot remember	73	14.1
Never	8	1.5
Did you receive a prescription? (n = 509) ^1^		
Yes	465	91.4
No	24	4.7
Cannot remember	20	3.9
Advice on taking antibiotics (n = 509) ^1^		
Yes, I received advice on how to take them	465	91.4
No	23	4.5
Cannot remember	21	4.1
Source of antibiotics (n = 509) ^1^		
Medical store or pharmacy	484	95.1
Friend or family member	5	1.0
I kept it from the previous time	5	1.0
Somewhere/someone else	5	1.0
Cannot remember	10	2.0

^1^ Only participants who reported using antibiotics were required to answer this question.

**Table 3 healthcare-11-02284-t003:** The association between the socio-demographics of the participants and their level of knowledge of antibiotics and antibiotic resistance.

	Knowledge of Antibiotics			Knowledge of Antibiotic Resistance		
Characteristic(s)	Low Knowledge, n (%)	High Knowledge, n (%)	X^2^ (*p*-Value)	Low Knowledge n (%)	High Knowledge n (%)	X^2^ (*p*-Value)
Gender						
Male	72 (43.4)	94 (56.6)	0.477 (0.490)	51 (30.7)	115 (69.3)	2.930 (0.087) ^1^
Female	141 (40.2)	210 (59.8)		135 (38.5)	216 (61.5)	
Age						
18–29	90 (60.8)	58 (39.2)	33.464 (0.000) ^1^	61 (41.2)	87 (58.8)	5.457 (0.065) ^1^
30–49	76 (31.9)	162 (68.1)		73 (30.7)	165 (69.3)	
50–60+	47 (35.9)	84 (64.1)		52 (39.7)	79 (60.3)	
Race						
Malay	61 (39.9)	92 (60.1)	16.777 (0.001) ^1^	49 (32.0)	104 (68.0)	15.495 (0.001) ^1^
Chinese	45 (29.4)	108 (70.6)		42 (27.5)	111 (72.5)	
Indian	87 (50.9)	84 (49.1)		73 (42.7)	98 (57.3)	
Other	20 (50.0)	20 (50.0)		22 (55.0)	18 (45.0)	
Household Income						
Low Income	93 (49.2)	96 (50.8)	9.700 (0.008) ^1^	93 (49.2)	96 (50.8)	24.129 (0.000) ^1^
Average Income	74 (39.8)	112 (60.2)		58 (31.2)	128 (68.8)	
Rich Income	46 (32.4)	96 (67.6)		35 (24.6)	107 (75.4)	
Education Level						
High school and less	24 (70.6)	10 (29.4)	19.450 (0.000) ^1^	26 (76.5)	8 (23.5)	32.816 (0.000) ^1^
Post-Secondary	58 (49.2)	60 (50.8)		51 (43.2)	67 (56.8)	
Higher Education	131 (35.9)	234 (64.1)		109 (29.9)	256 (70.1)	
Region						
Northern Region	28 (39.4)	43 (60.6)	6.736 (0.150)	33 (46.5)	38 (53.5)	12.114 (0.015) ^1^
Southern Region	8 (21.6)	29 (78.4)		5 (13.5)	32 (86.5)	
Central Region	157 (43.5)	204 (56.5)		133 (36.8)	228 (63.2)	
East Coast	3 (42.9)	4 (57.1)		2 (28.6)	5 (71.4)	
Borneo Island	17 (41.5)	24 (58.5)		13 (31.7)	28 (68.3)	
Settlement Category						
Urban	152 (41.3)	216 (58.7)	1.740 (0.419)	133 (36.1)	235 (63.9)	3.343 (0.188)
Suburban	52 (39.1)	81 (60.9)		44 (33.1)	89 (66.9)	
Rural	9 (56.3)	7 (43.8)		9 (56.3)	7 (43.8)	
Household Composition						
Adults only	133 (42.9)	177 (57.1)	0.928 (0.335)	119 (38.4)	191 (61.6)	1.953 (0.162)
Adult and children	80 (38.6)	127 (61.4)		67 (32.4)	140 (67.6)	

^1^ Parameters with *p* < 0.1 were selected for the multivariate logistic regression.

**Table 4 healthcare-11-02284-t004:** Odds ratio of having a better knowledge of antibiotics and antibiotic resistance.

	Knowledge of Antibiotics ^1^					Knowledge of Antibiotic Resistance ^1^				
Characteristic	B	*p*-Value	Unadjusted OR	Adjusted OR ^2^	CI Adjusted OR	B	*p*-Value	Unadjusted OR	Adjusted OR ^2^	CI Adjusted OR
Gender ^	0.132	0.490	1.141	1.302	0.874–1.939	−0.343	0.088	0.710	0.788	0.518–1.200
Age	0.548	0.000	1.730	1.620	1.253–2.095	0.045	0.719	1.046	0.876	0.674–1.140
Race	−0.228	0.016	0.796	0.791	0.650–0.961	−0.308	0.002	0.735	0.692	0.564–0.848
Household Income	0.354	0.002	1.425	1.126	0.872–1.453	0.567	0.000	1.764	1.495	1.143–1.954
Education Level	0.648	0.000	1.912	1.751	1.260–2.433	0.820	0.000	2.270	2.038	1.455–2.855
Region ^	−0.088	0.349	0.916	0.998	0.823–1.211	0.097	0.311	1.102	1.255	1.025–1.537

^1^ B = unstandardized coefficient; OR = odds ratio; CI = confidence interval. ^2^ Adjusted OR is calculated based on the multivariate model. Factors marked ^ are only included in the multivariate model for “Knowledge of Antibiotic Resistance”.

**Table 5 healthcare-11-02284-t005:** Correlations between knowledge of antibiotics and knowledge of antibiotic resistance.

Variables	Correlation Coefficient	*p*-Value
Knowledge of Antibiotics– Knowledge of Antibiotic Resistance	0.306	0.000

## Data Availability

The data presented in this study are available in the original article and Appendix A.

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
