# Peer review of "Socio-Demographic Factors and Public Knowledge of Antibiotic Resistance"

_healthcare, 2023, doi:10.3390/healthcare11162284_

Round 1

Reviewer 1 Report

Dear authors

The article must be improved and revised before publication.

The article needs to be checked for English.

The abstract is insufficiently elaborated. Mainly the results part should be more comprehensive and the conclusion part, even one sentence is enough.

The work lacks a comparison of the results with public awareness in other Asian countries or in the world.

The discussion is insufficiently elaborated. In the introduction and the discussion, sources regarding the occurrence of resistance and the consumption of individual antibiotics in Malaysia should be mentioned. Compare it with the world and Asian consumption of antibiotics and connect it with the knowledge in the given country.

362-369 This section needs to be reformulated and shortened because it lowers the level of results achieved in the given study.

In part conclusion it was found significant plagiarisms, therefore it is necessary to change formulations.

Best regards

The article needs to be checked for English.

Reviewer 2 Report

The authors have correctly identified the drawback of the study which they have written as “Due to the online nature of the questionnaire, it is impossible to fully monitor the 366 truthfulness and understanding of the respondents”. If the authors were aware of this, why didn’t they use in person survey? Many of the respondents may have googled the questions and marked correct answers!

The introduction part of unnecessarily lengthy and some of the sentences such as “There are also indirect costs such as the low productivity of the labor force because of an increase in mortality rate due to AMR” seem exaggerated for the time being. Common infections in the labor force leading to morbidity and absence from work are understandable but death!!!

Why was ‘sore-throat’ labeled as the wrong choice? A sore throat could be due to infection and viral or bacterial throat infections are usually described by the common people as sore throat!

Reviewer 3 Report

The manuscript assesses the knowledge of antibiotic use and antibiotic resistance in five regions of Malaysia.

General comments:

A laborious manuscript. However, some statements should be better explained: 1) is this survey based on WHO methodology, and what are the inclusion and exclusion criteria (?); 2) the questionnaire in the supplementary table should be updated, including the sections and the question numbers described in item 2.2. (survey Questionary); 3) The part of the questionnaire concerning “antibiotic use in Agriculture” can be considered quite restrictive, just three specific questions (a. Yes; b. No; c. Don’t know) directed at rural people. I think these results do not add value to the present work. 4)  were the antibiotics obtained at the medical store or pharmacy without a doctor's prescription? 5) In the item Discussion, it has been highlighted that a possible reason for this misconception could be due to physicians who unnecessarily prescribe antibiotics out of patients’ expectations; reductive this conclusion, I propose to discuss this point taking into account doctor’s attitudes and the intrinsic/extrinsic factors (please see reference). 6) This study assesses the knowledge of antibiotics and antibiotic resistance among the public in Malaysia to determine the gaps in knowledge that exist. I propose the authors to improve this sentence.

Specific Comments:

L. 98: 18 years old

L. 186: replace okay with the term appropriate

Reviewer 4 Report

In their paper titled: “Knowledge of Antibiotic Use and Antibiotic Resistance: Findings from a Nationwide Survey in Malaysia”, the authors survey the Malaysian population about their knowledge of antibiotic resistance. The paper finds that young and low-income populations lack knowledge, compared to the advanced and educated population.  The paper is well-written, the statistics look fine, and I have only a few comments.

Major comments:

-Associating knowledge (and lack thereof) with ethnicity can potentially lead to ethnic prejudice and racial intolerance. I don’t know about Malaysia, but in the US and Europe, a claim that this or that ethnicity is inferior is considered racism.  Therefore, please erase the sentence, “Chinese ethnicity (X2 = 16.777, p=0.001)”(line 177). “Chinese 210 ethnicity,”(line 210). Also, the ethnicities mentioned in Tables 3 and 4 should be deleted.

-The paper ignores that antibiotic resistance is ancient, and predates modern medicine, by millions of years. Please add a sentence acknowledging this fact, in the introduction.  For example, you could write: “However, throughout evolution, bacteria have developed resistance towards antibiotics, causing the medication to be ineffective in fighting infections (D'Costa VM, King CE, Kalan L, et al. Antibiotic resistance is ancient. Nature 2011;477:457-61)”

-The paper ignores the cyclic nature of antibiotic resistance, which rises and falls with use and disuse, respectively. Please add a sentence acknowledging this fact in the introduction.  For example, you could write: ”Antibiotic resistance rises and falls in a cyclic manner, increasing with antibiotic use, and decreasing with its disuse (Dabour R, Meirson T, and Samson AO Global antibiotic resistance is mostly periodic 2016, J Glob Antimicrob Resist. 7:132-134)”.

-The paper clearly shows the lack of knowledge in the young and poor, and exposes a serious problem in Malaysia, which is likely shared with the rest of the world.  The data is compelling, and the paper is interesting to read. However, one question remains unanswered: “What did we do to solve the problem?”.  One simple solution is to get in touch with the survey responders, and inform them about antibiotic resistance (if you can contact them?). Another solution is that the authors post a 2-3 minutes YouTube video in Bahasa Melayu about antibiotic resistance and its use. A third solution is to prepare a short lecture for laymen about antibiotic resistance and present it at social gatherings at University, in family gatherings, etc. Finally, the authors should write 2-3 sentences about their solution of choice in the paper discussion.  

Minor comments:

-The title is not specific. Please suggest a better title that highlights the main finding. Below is a suggestion, but you are welcome to propose another title: “Nationwide Survey in Malaysia Finds Deficient Knowledge of Antibiotic Resistance and its Use (particularly among young and low-income population?)”

-In the abstract, please correct: “A weak positive correlation was observed …”

-In p.2 line 56, please correct: “…have also been on the rise”

-On line 142, “There was a 142 higher representation of females (n= 351, 67.9%)”. This is not a criticism, just a request for clarification. Why do you think more women answered than men?

-Figure 2. In English the word physician is used, as Doctor is unspecific. Please change.

Reviewer 5 Report

You need to identify your rationale behind some of these methodologies. It is a well written paper, and like many I have reviewed in past, therefore there is nothing original here but it sheds data for developing your country's NAP.

Round 2

Reviewer 2 Report

Accept 

Reviewer 3 Report

The manuscript can be accept in present form